# From Industrial Food Waste to Bioactive Ingredients: A Review on the Sustainable Management and Transformation of Plant-Derived Food Waste

**DOI:** 10.3390/foods12112183

**Published:** 2023-05-29

**Authors:** Yassine Jaouhari, F. Travaglia, L. Giovannelli, A. Picco, E. Oz, F. Oz, M. Bordiga

**Affiliations:** 1Department of Pharmaceutical Sciences, Università degli Studi del Piemonte Orientale “A. Avogadro”, Largo Donegani 2, 28100 Novara, Italy; yassine.jaouhari@uniupo.it (Y.J.); fabiano.travaglia@uniupo.it (F.T.); lorella.giovannelli@uniupo.it (L.G.);; 2Department of Food Engineering, Faculty of Agriculture, Ataturk University, Erzurum 25240, Türkiye; emel.oz@atauni.edu.tr

**Keywords:** food waste, valorization, sustainability, circular economy, nutraceuticals, cosmetics

## Abstract

According to the United Nations, approximately one-third of the food produced for human consumption is wasted. The actual linear “Take-Make-Dispose” model is nowadays obsolete and uneconomical for societies and the environment, while circular thinking in production systems and its effective adoption offers new opportunities and benefits. Following the “Waste Framework Directive” (2008/98/CE), the European Green Deal, and the actual Circular Economy Action Plan, when prevention is not possible, recovering an unavoidable food waste as a by-product represents a most promising pathway. Using last year’s by-products, which are rich in nutrients and bioactive compounds, such as dietary fiber, polyphenols, and peptides, offer a wake-up call to the nutraceutical and cosmetic industry to invest and develop value-added products generated from food waste ingredients.

## 1. Introduction

Through adopting the first Circular Economy Action Plan in 2015, food waste was identified as a priority area by the European Commission for global sustainable development [1].

Among the 17 Sustainable Development Goals (SDGs), which were defined by the UN 2030 Agenda signed in 2015, the SDG-12 (Responsible Consumption and Production) and its targets has become a major challenge as the amount of industrial food waste in the world continues to increase [2].

The SDG’s present a golden opportunity to convert the abstract idea of sustainability into an action plan, which must be linked to a sense of urgency, in order to achieve specific objectives to shift the world onto a sustainable and resilient path.

The main objective of the SDG-12 is to halve worldwide food waste and food loss by 2030. Reducing food loss and waste also contributes to other SDGs, including the Climate Change (SDG-13), Zero Hunger (SDG-2), Sustainable Water Management (SDG-6), and Biodiversity and Terrestrial Ecosystems (SDG-15).

According to the most recent “Food Waste Index Report” published by UNEP (United Nations Environment Programme), 931 million tons of food waste were produced in 2021 across all stages of food supply chain: production, distribution, retail, markets, and domestic consumption [3]. This report aims to support the SDG-12 and generate a new estimate every year.

Moreover, food waste was a crucial part of the conversation at side events surrounding the “26th UN Climate Change Conference” (COP26) held in November 2021, which took place at the Scottish Event Campus (SEC) in Glasgow [4].

Food waste can be classified into pre-consumer waste, manufacturing waste, and post-consumer waste [5]. Manufacturing or industrial food waste products are usually organic residues defined as unavoidable food waste produced during sorting and cleaning processes and are essentially represented by plant-derived waste, such as fruit and vegetable peels or pomace; grains and cereals bran; and animal-derived waste that have no economic value for the manufacturing company, thus being destined to be treated according to the linear model [6]. On the other hand, if these waste products are intended to be reused by the same company or other organizations, they cannot be considered waste but instead become a by-product, often with a market value [7].

This novel approach based on circular thinking can provide the correct solutions to emerging technical, economic, and environmental challenges [8].

Agri-food waste generated via industrial processes will be redirected to be used as a “new” input in another process by one or several organizations, in particular universities, start-up firms, and small companies, creating a mutual advantage that is pivotal for realizing the transition from the linear to the circular economy based on sustainable development [9].

Industrial food waste can be less visible to the public: as mentioned above, approximately 1 billion tons of food waste are annually produced, which roughly corresponds to one-third of the food produced for human consumption [10]. About 38% of this waste occurs during food processing (unavoidable food waste): the beverage industry produces about 26% of industrial food waste, followed by the ice cream industry (21.3%), fruit and vegetable production (14.8%), manufacture of grain and starch products (12.9%), and meat production (8%) [11]. 

In industrialized countries, food waste, which accounts for approximately 56% of global food waste, is obtained via the latter stages of the supply chain, particularly during processing and consumption [12]. In addition to the conventional waste represented by the residues obtained at the end of the manufacturing processes (e.g., pomace from pressing), more recently, the “culture for perfect foods” regarding the aesthetic standards of product’s weight, shape, and size contributed considerably to the increase in the amount of food waste. The waste generated in industrialized regions is related to the very demanding quality standards demanded by retailers and influenced by consumers’ choices [13]. 

In contrast, in developing countries, due to financial and technical shortages, most of this waste is obtained in the early stages, especially during harvesting and storage [14]. 

Efficient waste management and the ability to convert food waste into by-products in both industrialized and developing countries could greatly influence the so-called “three pillars of sustainability”—social, economic, and environmental—which are intertwined [15].

Many benefits can be obtained through reducing food waste and achieving the SDG-12: support the fight against climate change (environmental pillar), save money for companies (economical pillar), create job and boost local communities (social pillar), and eradicate malnutrition through producing novel products (social pillar) [2,16].

Today, the promotion of sustainability in the agri-food sector is driven by endogenous factors, such as the commitment of modern companies, and exogenous factors, such as consumer choice, which is increasingly focused on the issue of sustainability, as well as national and international regulations [17].

Beyond the introduction, this review is divided into three sections. Section one presents the role of the food industries in waste management in terms of sustainable development, corporate sustainability, and reporting and investment. Moreover, it investigates the impact of industrial food waste generation from environmental, economic, and social points of view. The second section introduces the current European legislation on food waste and by-products. Finally, section three highlights a summary of the recent literature about the valorization and utilization of plant-derived food waste at patent and industrial levels.

## 2. Sustainability and Food Waste: A Literature Review

### 2.1. Sustainable Development 

During the last two decades, food companies became protagonists in driving changes in their business models. One of the driving factors of these positive shifts was to satisfy the requests of internal and external stakeholders across the entire food chain [18]. A stakeholder is defined as a single person or group who/that has an interest in the company and can either affect or be affected by its business [19]. Stakeholders are commonly represented by consumers, customers, investors, employees, suppliers, labor unions, associations, and regulatory authorities; in the last few years, stakeholders showed, from an ethical point of view, particular interest in production processes and the related impacts toward the environment and society [20]. 

Companies operate more effectively when they build internal capacities to foster sustainability and make positive contributions to their stakeholders and the planet. Capacity building includes expanding sustainability awareness and knowledge, as well as fostering the skills and processes needed to create sustainable development. 

The concept of sustainable development is an idea that was formalized for the first time in 1987 by the World Commission on Environment and Development (WCED) in the “Brundtland Report”. The report defined sustainable development as a “development that allows the present generation to satisfy their own needs without compromising the possibility of future generations to satisfy their own.” [21]. The EU’s most prominent initiative to ensure sustainable development was a program entitled, “Towards Sustainability”, which was introduced in 1992, which aimed to encourage the idea that the environment and resources are not unlimited to future generations [22].

Actually, this novel trend is derived entirely from the fact that companies find themselves in a situation that makes it impossible for them to operate from a purely economic perspective. In this scenario, companies adapt their standard schemes to face social, environmental, and economic issues raised by their stakeholders, with the intention of making their processes more sustainable with few negative effects on future generations [23]. 

Sustainable development should be considered a universal goal for every business organization. According to the Triple Line Bottom concept described in Elkington’s book, “Cannibals with Forks: The Triple Bottom Line of 21st-Century Business”, sustainability is seen in terms of three dimensions defined as sustainability pillars: social, economic, and environmental pillars. The Triple Line Bottom was defined as follows: “Triple Bottom Line accounting attempts to describe the social and environmental impact of an organization’s activities, in a measurable way, to its economic performance in order to show improvement or to make evaluation more in-depth” [24]. Later, the same author attempted to clarify the pillars’ meaning by naming them in a novel way: Profit, Planet, and People [25]. Each dimension is intertwined with the others, and all pillars must work together to drive any sustainability effort. This concept represents an evolution of the conventional bottom line, where the conventional measure of corporate success is only derived from the business profit. This situation represents a big problem that still affects several companies, which find themselves trapped in an obsolete and outdated approach to value creation. Focused on improving short-term profits, they ignore the depletion of natural resources, the well-being of the workers, and vulnerability of the community around which the business revolves. As described by Kramer et al. (2011), the solution to reconcile economic profits and stakeholder expectations lies in the principle of “shared value”, which involves creating economic value in a way that also creates value for the community and society [26]. As it is still a complex topic, the concept of sustainable development could be interpreted at all decision-making levels. The first level is governments and global policies. The second level is companies that try to voluntarily improve their sustainability aspect, which is communicated through targets, strategy, and reports. The individual consumers who can improve their consumption choices represent the final level [27].

### 2.2. Corporate Social Responsibility

The modern commitment linked to sustainable development adopted by companies, governments, and non-governmental organizations takes the name of Corporate Social Responsibility (CSR), which has as its main objective the improvement of the well-being of communities through the use and application of targeted economic and social resources [28]. Long-term financial performance increases when CSR issues, such as sustainable development, work, and family reconciliation and professional ethics, become part of the business core. 

The basic idea of modern corporate social responsibility was developed during the first decade of the 20th century. The first CSR concept emerged in America in 1917 when the declines in individual and social ethics were the subject of notable debates aiming to increase awareness among managers and entrepreneurs of how to legitimize businesses for social and environmental improvement. Things have changed significantly since then [29]. 

In 1953, Howard Bowen, who is considered one of the pioneers of CSR, described this emergent aspect in the book, “Social Responsibilities of the Businessman” [30]. According to Bowen, business activities were considered the decision-making centers that influenced the lives of the community in many aspects. The same author defined CSR as an “obligation of businessmen to follow those policies, take those decisions or follow those action lines that are desired by the objectives and values of our society”. Actually, the most often-quoted definition came from Archie Carroll. His idea, which went beyond the mere idea of profit and of law obedience, was explained as “Corporate social responsibility of business encompasses the economic, legal, ethical and discretionary expectations that society has of organizations at a given point in time” [31]. 

Starting from this definition, the same author created a simple graphical representation of CSR in the form of a pyramid in order to meet the economic, legal, ethical, and philanthropic demands of business organizations [32] (Figure 1). 

Carroll’s pyramid represented the building block of socially responsible business practices, which are as follows: to be economically profitable, to obey the law, to be ethically responsible, and to donate to philanthropic causes. On the same track, the Commission of the European Communities (2002) defined CSR as follows: “a concept whereby companies integrate social and environmental concerns in their business operations and in their interaction with their stakeholders on a voluntary basis” [33].

More explicitly, Carter and Jennings (2004) indicated that CSR is not only synonymous with business ethics, but also encompasses other dimensions, including diversity, safety, and environment [34]. A more recent definition was proposed by Stobierski (2021), who defined CSR as a concept where modern business has a responsibility to the society that exists around it, and often this concept is divided into four types of responsibility: environmental, philanthropic, ethical, and economic [35].

With this in mind, CSR should be including purely voluntary actions to be considered socially responsible, and rules for righteousness need to come from human empathy and ethics, without forgetting the economic responsibility to the society that permitted them to be created and sustained. 

In recent years, an emergent aspect of CSR was the integration of environmental management awareness into a business’s mentality and activities in order to promote the transformation of manufacturing into green manufacturing in the short term [36]. 

Furthermore, environmental CSR is an important part of the concept of CSR. Due to its complexity, environmental issues also require spontaneous co-operation and voluntary participation carried out by managers and employees beyond work requirements [37]. Environmental CSR is a strategic guideline for companies in achieving sustainable development goals [38], and the actual literature can define it from three perspectives: firstly, based on the series of environmental measures and approaches that companies voluntarily take [39]. The second process-based view of environmental CSR is defined as a business optimization process designed to reduce its impact on the environment. The final perspective considers how companies provide eco-friendly services [40]. In other words, the actual business model is committed to incorporating environmentalism into routine business operations.

Moreover, a growing number of companies known for their environmental initiatives, such as waste reduction, recycling, and use of renewable energy, previously made important efforts to improve their sustainable performance. Examples of environment-oriented CSR in the food chain include the case of a English coffee shop chain that offers discounts to their customers if they bring a reusable cup in order to reduce plastic coffee cups waste [41].

Overall, CSR is a strategy employed by businesses not only to increase profits, but to also play an active and constructive social role in the environment around them. Through their real positive difference, such organizations make a significant impact in enhancing their company’s reputation globally, as well as playing a critical part in boosting customer confidence. 

### 2.3. Sustainability Strategies

#### 2.3.1. Stakeholder Engagement

Defining, measuring, managing, and communicating sustainability became fundamental steps in pursuing a corporate strategy. The critical points for the company that pursues a sustainable strategy are derived from the difficulty of measuring and correctly interpreting the effects of its actions from economic, environmental, and social points of view.

Sustainable development requires voluntary approaches and largely depends on corporate philosophy and transparency toward stakeholders [18,42,43]. 

The first step is to recognize activities and issues that matter most to the company’s stakeholders and analyze how economic, social, and environmental impacts are perceived along the value chain. These preliminary activities are defined as “materiality assessments”, which are strategic business tools aimed at engaging stakeholders to identify and prioritize what they want to measure [44]. Materiality assessments allow organizations to bring out more clearly the relationships between their corporate interests and those of the stakeholders. The materiality analysis is a structured process that typically involves gathering information through the assessment of questionnaires and initiatives with stakeholders (external engagement) and the company’s spokespeople (internal engagement) in order to define new emerging sustainable themes [45]. The importance and impact of each theme identified is rated on a numerical scale. This method will give quantitative data that can be analyzed and visually explained. Relevant themes are graphically prioritized onto a materiality matrix, which is a typical tool used to understand new sustainable trends and the economic, social, and environmental impact of corporate activities [46] (Figure 2). Each material topic identified has a particular impact, which is directly generated by the organization and directly perceived by the external stakeholders. 

The materiality determination process reflects the transparency of the company and is an important part of sustainability reporting and disclosure. Once the most material environmental, social, and governance (ESG) issues are identified, the organization can develop strategies and action plans to address these issues, set targets and goals for improvement, and report progress to stakeholders [47].

#### 2.3.2. Targets, Key Performance Indicators and Reporting

A materiality assessment is the first step in setting ESG targets in an industrial context. Setting sustainability targets is important because it provides a clear roadmap for progress towards sustainable development. Targets are the backbone of an effective sustainability strategy and refer to the practice of setting specific and measurable goals related to sustainable development and environmental responsibility, before working towards achieving those goals over a defined period. These targets can be set by businesses, organizations, governments, or individuals, and are often informed by frameworks such as the United Nations Sustainable Development Goals (SDGs) [48].

In general, good targets should be measurable, material, complete, consistent, and ambitious [49]. Examples of sustainability targets may include achieving a certain level of renewable energy use, implementing sustainable supply chain practices, reducing waste and water usage, and improving welfare and well-being practices.

Once the corporate targets and the consequent actions are defined, the most complex part concerns the selection of appropriate indicators for monitoring performance in relation to the objectives and performance levels that the company wants to achieve. According to Alshehhi et al., sustainability and performance are related aspects of the success of a company [50].

Through developing sustainability strategies with specific targets that can be measured and converted into Key Performances Indicators (KPIs), companies can also fulfill transparency requirements. Peter Drucker, who was one of the great management theorists, stated the idea “You can’t manage what you can’t measure”, suggesting that in order to effectively manage a process, it is necessary to have some way of quantifying or measuring progress towards the desired outcomes [51]. This concept is often applied in the context of KPIs, which represent a quantifiable measurement used to monitor and track sustainability performance towards a target. These indicators can be used to measure a wide range of business activities, especially in the agri-food system, and can provide valuable insights into business performance. Overall, sustainability indicators aim to translate the key sustainable development issues for the industry into relevant measures related to sustainability performance [52]. For this reason, a good KPI should be clearly defined and measurable, tied to the business objectives, realistic and achievable, set within a specific timeframe, and finally, it must provide insights that can be used to make informed decisions [53]. According to Singh et al. (2009), KPIs are an important tool that “simplify, quantify, analyze information that would otherwise be complex and complicated, making it a manageable and significant quantity” [54].

Companies that have always been characterized based on the goal of maintaining steady their social, environmental and economic balances need to annually report their sustainability practices to their stakeholders [55]. As reiterated by Junior et al. (2014), disclosing fundamental aspects of sustainability reporting processes enhances transparency about organizational performance [56]. Moreover, reporting is used to change public opinion regarding the company into a realistic view. Today, sustainability reporting can decrease the so-called “information asymmetries” between companies and relevant stakeholders [57]. According to Daub et al. (2007), a sustainability report is a report that “must contain qualitative and quantitative information on the extent to which the company has managed to improve its economic, environmental and social effectiveness and efficiency in the reporting period and integrate these aspects in a sustainability management system.” [58].

However, it must be emphasized that choosing the right indicators to report corporate improvement is very challenging due to a lack of a standardized set of both KPIs and sustainability reporting rules [59]. Over time, frameworks and directives were designed as guidelines for achieving corporate sustainability.

The year 2014 was a defining period for corporate non-financial reporting in Europe. The introduction of the EU Directive 2014/95/EU on the disclosure of Non-Financial Information (NFI) set a clear course toward transparency and accountability on social and environmental issues [60]. This directive was a milestone and harmonization principle for companies, who are now requested to include relevant NFI in their sustainability annual report. Moreover, the new directive plays an important role in Europe’s commitment to meeting the United Nations Sustainable Development Goals (SDGs). 

Previous studies compared sustainability reports issued before and after the EU Directive. Most of them found a significant increase in the quality and quantity of disclosure [61,62,63]. 

As a result of this regulation and the need to comply with all aspects requested by the EU on the disclosure of non-financial and diversity information, several companies started to adopt guidelines and standardized KPIs frameworks [64].

Given the scenario, the Global Reporting Initiative (GRI) stood out in recent years and remains the most commonly used framework that provides a set of KPIs for sustainability reporting, which can be adopted on a global scale by companies and non-corporate organizations [65,66]. GRI is an independent international organization that has pioneered corporate sustainability reporting since 1997. The GRI framework also includes guidance on how to report the provided indicators and the provision of contextual information to help the stakeholders to understand and interpret the organization’s sustainability performance.

Several studies highlighted that the GRI guidelines became the common standards followed in different countries [67,68]. Roca et al. (2011) found that almost half of the examined corporate reports state that they use the GRI guidelines [69]. 

The GRI Sustainability Reporting Standards place emphasis on sustainability matters that consist of a set of 36 interconnected standards applicable to all organizations. This set includes a modular system of indicators, which are organized into three universal standards (GRI 101: Foundation, GRI 102: General Disclosures, and GRI 103: Management Approach) and 33 topic-specific standards, as well as divided into Economic (GRI 200), Environmental (GRI 300), and Social (GRI 400) series. Each standard contains disclosure topics and KPIs, which, in turn, enhance an organization’s engagement with the SDGs through mapping each of the 17 goals [70].

Considering the food waste themes present in the agri-food sector, the Environmental GRI topic GRI 306 (waste) includes specific disclosures and KPIs to report companies’ impacts and performances. These guidelines provide insight into the level of control that the organization assumes for waste management, waste generation, and use of recycled input materials [71].

#### 2.3.3. Investments

The transition towards a circular and sustainable model requires voluntary approaches, and the principal strategy is represented by substantial investments [72]. A good management of economic resources contributes to the development of the entire company and the community. This thinking largely depends on corporate philosophy and transparency toward the stakeholders.

Investments in human capital are essential to build innovative solutions since this resource is closely related to the company’s productive capacity. Staff training, improving internal communication, promoting gender equality, and boosting employee wellbeing represent investments in human capital by modern economies that are focused on innovation, quality, and creativity [73]. The resulting employee satisfaction is a powerful performance engine for the company: the motivation and belongingness represent a starting point for building stable teams and achieving sustainability goals. 

In parallel, economic investments in green technological innovation are very important in order to minimize, for instance, energy consumption and pollutant waste. As mentioned by De Mello et al. (2008), innovation is defined as “a key element of corporate competitiveness in the 21st century, and has therefore attracted special attention of management researchers and practitioners” [74]. Nowadays, innovation is a fundamental element for companies as it affects profits and the environment and allows them to respond quickly to changes and challenges. 

Sustainable innovation in specific companies’ areas, such as R&D and production plants, can contribute to a more efficient future through stimulating the growth and development of targets synonymous with corporate sustainability: digitalization, decarbonization, and the circular economy [75].

Over time, the business practices in the agri-food system focused solely on profit ignored serious problems related to their production processes. 

In reality, the environmental and climatic issue was always of considerable importance. Hristov et al. (2019) suggest that waste issues represent the principal environmental KPIs that influence the creation of value [76].

Companies have to deal with related aspects of waste generation, such as climate change and pollution, regardless of where they are located and to which business sector they belong. All waste generated carries valuable and sometimes limited resources previously invested in the food supply chain, such as water, fuels, and labor costs.

### 2.4. Impact of Food Waste from an Environmental, Economic, and Social Point of View

#### 2.4.1. Environmental Impacts

A global research study published by Paul Hawken ranked food waste reduction as the 3rd most important of 100 solutions required to reduce climate change [77].

Currently, a large part of all food wastage undergoes disposal treatments, while only a low percentage is reused or recycled [78].

Food waste treatment and disposal are carried out primarily via one of these methods: dumps and landfills, thermal treatment (incineration, gasification, and pyrolysis, open burning), and biological waste treatment (composting and anaerobic digestion) [79].

Having said that, it is easy to understand the way that food waste affects climate change and food security in various ways. In fact, food waste represents a potentially significant source of global greenhouse gas (GHG) emissions, which are released during its treatment, and a territorial risk when they end up in landfills or illegal dumpsites.

Until recently, the carbon footprint of food waste decomposition and incineration was estimated at 3.3 billion tons of CO_2_ equivalent per year, which corresponds to 8–10% of global greenhouse gas emissions [80]. 

Food waste could contain complex substances, such as heavy metals, mycotoxins, and other toxicants, that represent a source of secondary contamination of the adjacent environments. Organic waste stored in landfills can degrade and contaminate the soil, the surface, and groundwater, contributing in several ways to the chemical and biological transformation of the environment and the loss of land and water biodiversity.

In this regard, Mosina et al. studied the agro-ecological and geochemical characteristics of the contaminated soil with food waste. The number of microscopic fungi in the contaminated soil increased seven times, and the phytotoxic fungi of the genus Alternaria were found in pea seedlings (*Pisum sativum* L.). Moreover, soil contamination is linked to the alteration of the species composition of the microbial population in the soil [81].

Geographically fruits and vegetables waste generates a substantial impact on the environment, in terms of carbon footprint during their treatment, in industrialized Asia (China, Japan, Republic of Korea), Europe, and Latin America, while wastage of cereals, especially straw and bran rice, represents a vicious cycle of rice production and waste incineration in Asian countries [82].

#### 2.4.2. Economic Impacts

Besides the climatic aspect, it is also necessary to consider the economic cost for the companies to treat food waste: incineration possesses the lowest life cycle costs (LCC) (80 euro/ton), while landfilling is the most expensive option (140 euro/ton) [83]. 

Globally, the financial costs of food wastage treatment amount to approximately 1 trillion dollars (US) each year (FAO), which is roughly equivalent to Spain’s gross domestic product, without considering the costs related to environmental and social impacts. Environmental costs reach around 700 billion and social costs reach around 900 billion dollars; these costs include loss of livelihoods due to soil erosion, risks to biodiversity, and greenhouse emissions [84].

In a closed-loop system, wastes are not disposed of and continue to be of value without incurring costs for their management.

A report from McKinsey suggested that, in Europe alone, investing and adopting circular economy principles could generate a net economic benefit of 1.8 trillion euros for manufacturing companies by 2030 [85].

Industries’ investments and partnerships that reduce food loss and waste can improve food security, minimize greenhouse gas emissions, and save money.

Regarding the latter aspect, a global study indicates that companies could save 14 dollars or more in operating costs for every 1 dollar invested in reducing food waste [86]. Moreover, European companies in the agrifood sector could save up to 10 billion euros annually through introducing innovations that reduce waste in the food chain [87].

According to the “Food Waste Capital Tracker” database published by Rethinking Food Waste Through Economics and Data (ReFED), which analyzed various economic inputs related to waste management, in 2021, total private investment in food waste technology solutions reached 1.9 billion dollars [88].

Companies’ investments may include quantifying and monitoring food loss and waste, training staff on practices that reduce waste, changing food storage and handling processes, changing packaging to extend shelf-life, and collaborating with experts employed by universities and research centers. 

Moreover, if well managed, the circular economy presents an opportunity to enhance “green” job creation. The expansion of “reuse and remanufacturing” could create up to 3 million extra jobs in the European Union by 2030, which is three times more than in the actual business scenario [89].

Starting from 2014, European Commission examined 21 companies that adopted circular economy measures and showed that all these companies witnessed net job creation. Net job creation ranged from 1.3% for some larger companies to 8.4% for smaller ones [90].

#### 2.4.3. Social Impacts

Social impacts of food waste and its non-reuse may be attributed to an ethical and moral dimension within the general concept of global food security, as more than 800 million people across the globe suffer from hunger [91]. For each kind of food waste characterized, a set of waste management approaches are suggested in order to minimize environmental impacts and maximize social and economic benefits.

Although the best option is the prevention of food waste production, according to the latest European management directive, the valorization of food waste can be considered one of the best methods [92].

In order to understand how these challenges can be transformed into opportunities, the food waste composition should be clearly characterized.

Food wastes and by-products represent a source of high-value-added compounds, such as proteins, fiber, oligosaccharides, fatty acids, lipids, polyphenols, food-grade pigments, and terpenes, that could be exploited in different industrial sectors [93,94,95]. These compounds are naturally present in edible and non-edible parts, and the second part often contains higher amounts of bioactive compounds, such as fiber and oligosaccharides, when compared to the edible parts [96]. 

The utilization of residues generated in the food manufacturing processes located in poor regions could be a crucial strategy to fight malnutrition through the formulation of novel foods or food fortification that will directly benefit local communities [97].

In summary, the recovery of these compounds from fruits and vegetables peels and pomace or bran cereals to produce value-added products for the nutraceutical and cosmetic applications, which occurs through exploiting the principles of green chemistry, can provide a sustainable basis for industries and stimulate research development.

The creation of an innovative methodology for the extraction, purification, quantification, and stabilization of biochemicals, which are usually discarded as waste, is also considered a crucial step for sustainable development [98]. 

Production of nutraceuticals and cosmetics from waste reduces the amount of material that would otherwise be sent to landfills and, in addition, can generate a marketable commodity.

## 3. Current European Strategy and Legislation on Food Waste and By-Product

While prevention is not possible, converting unavoidable food waste into a by-product represents the most promising pathway to achieve the zero-waste goal and accelerate the transition of the food industry to a sustainable and circular economy. In a general context, food waste is referred to as food residual or edible residual material that is no longer useful or necessary. By-products, on the other hand, refer to residues generated during a manufacturing process; thus, they are not the main products but still have an economical and nutritional value [7]. While waste is typically considered a negative outcome of a process, by-products can have a positive impact on the food supply chain. The importance of focusing on and describing waste as a resource is promoted in contemporary studies that reinforce the concept of the circular economy [99,100]. 

Today, in order for food waste to achieve recognition as a by-product, several strict criteria must be applied. 

European Union rules on waste and by-product operate under Directive 2008/98 [92]. This Directive, which is better known as the “Waste Framework Directive”, is a legislative landmark in the area of waste in the European Union. The scope of this directive, which was adopted on 19 November 2008, is to establish a legal framework for preventing waste in the EU and protecting public health and the environment through promoting a package of measures based on circular economy principles. One of the key concepts in the Waste Framework Directive is the distinction between waste and by-products; the Directive defines all waste or food waste as “any substance or object which the holder discards or intends or is required to discard”. Fixing a consistent definition of waste has been a long-debated issue in the European Commission [101]. According to the Framework and, more precisely, Article 3 (15), under certain conditions, food waste may “cease to be waste” escaping the legal regime of waste. Any substance resulting from a production process is considered by-product if it meets the following criteria: -it is produced unintentionally during a production process;-it can be used directly without any further processing, other than normal industrial practice;-its further use is certain and meets environmental and health protection requirements.


While the first and second criteria can be easily achieved, primarily as a result of historical use, the quality standard of the by-product was not described because it may be impossible to establish generical uniform standards of quality. However, the absence of pathogens and contaminants, lower moisture content, and nutrient and bioactive compound availability should represent a general quality descriptor [102]. 

Indeed, according to the five steps of the “waste hierarchy” proposed in the same Directive, preventing waste is the preferred option, and sending waste to landfills should be the last resort.

The EU Platform on Food Loss and Food Waste in 2018, which was published following the actual Waste Directive, introduced obligations and measures for the Member States to prevent waste generation and loss [103]. The regulation’s target is reducing wastage by 50% within production and supply chains by 2030 as a contribution to meeting the United Nations Sustainable Development Goals.

Moreover, additional obligations are the monitoring and annual reporting on food waste levels. According to the relative provisions in the amended Waste Framework Directive, each Member State shall monitor the implementation of their prevention measure by reporting the level of food waste, thus favoring the generation of by-products rather than waste. 

Overall, the Waste Framework Directive aims to promote a circular economy in which waste is minimized, resources are conserved, and environmental protection is prioritized.

A recent breakthrough regarding the prevention and reduction in food waste is the release of two important policy documents: the European Green Deal (EGD) and the Circular Economy Action Plan (CEAP). These actions are both initiatives instigated by the European Commission that are aimed at promoting sustainability and reducing environmental impact. The EGD is a framework document adopted in 2019 with the overarching aim of making the European Union climate neutral by 2050 and halting biodiversity loss through initiatives in various sectors, including food waste [104]. The second document is the CEAP, which was approved in 2020 and described by the European Commission as “one of the main building blocks of the European Green Deal” [105]. The CEAP recognizes the significant impact of food waste on the environment and society and aims to tackle it through various measures. With a focus on sustainable agriculture and food production, the CEAP strategy foreseen in the European Green Deal proposes a range of actions, including targets to reduce food waste and the transition through innovation and digitalization. The European Commission proposed a target for food waste reduction as a key action under the forthcoming EU Farm-to-Fork Strategy [106]. The Farm-to-Fork Strategy recognizes the inextricable links between a sustainable food system and healthy people, healthy societies, and a healthy planet. Starting in 2022, it is expected to provide the first data regarding food waste generation in Member States in order to set a baseline and propose binding targets to reduce food waste.

Through smart specializations, such as LIFE and Horizon Europe, the Action Plan encourages innovation to reduce food waste and improve resource efficiency. This approach includes supporting research into and development of new technologies and business models with the aim of creating solutions for European businesses [107,108].

## 4. Recent Patents and Industrial Valorization of Vegetable By-Products for Food, Nutraceutical, and Cosmetic Applications

Few published works deal with the conversion of food waste extracts into raw materials suitable to be formulated into nutraceuticals or cosmetic products [109,110]. However, this scientific literature is reported at a laboratory level. The scale-up of the extraction processes remains a critical but often demanding step for the development of value-added products. Several studies on the industrial applications of upcycled food waste are ongoing, and new patents are gradually spreading. 

There is a wide range of available by-products stemming from different food industrial sectors that can be found in both the vegetable and animal worlds [111,112]. 

Nowadays, as mentioned in previous chapters, it is essential to adopt a circular economy approach, in which waste from a production process can prove to be an input for a new one. Managing these waste materials derived from fruits, vegetables, and cereals and making them exploitable in cosmetic and nutraceutical applications could represent a valuable and green strategy [113] (Figure 3). 

In this field, as described in Table 1 and in the following chapters, many patents were previously launched and many innovations were previously implemented for food, nutraceutical, and cosmetic applications.

### 4.1. Food & Nutraceutical Application

Today, research into the potential of using plant by-products in the food industry is gaining momentum, with the aim being creating new value-added products and managing waste in line with the goal of sustainability [129]. In addition, the use of vegetable by-products in the food industry offers a real opportunity to improve human health through the reformulation of processed foods as part of the FAO concept of “Ensure healthy lives and promote well-being for all at all ages” [84,130]. 

Bakery products are a food group with a high global consumption rate, and the main ingredient used in their production process is usually white wheat flour. White flour has low antioxidant activity and high nutritional value [131]. The inclusion of fruit and vegetable by-products in bakery products, such as bread, snacks, and biscuits, is a very popular phenomenon leveraged in research to improve functional properties [132,133]. Many fruit and vegetable by-products, such as carrot peels, beetroot, raspberry, and cranberry pomace, are also known to be effective in increasing the phenolic and dietary fiber content in bakery goods [134,135,136]. Since 2019, an American food company saved 79,000 kg of pressed vegetables from juice manufacturing. This company, which launched in 2015, makes snacks and chips using fresh vegetable juice and pulp, such as carrots, cucumbers, kale, and celery rich in dietary fiber and minerals [114]. 

The reuse of spent grains, such as barley, rye and wheat, and their by-products, such as germ or outer layers, represents a new frontier for modern food companies that intend to develop value-added products, especially in the bakery sector, in an innovative and sustainable way [137]. Grain by-products are sourced primarily from grain processors and the brewing industry. Cereal by-products, especially the bran which surrounds the endosperm, are rich in soluble and insoluble dietary fiber, such as cellulose, hemicellulose (β-glucans and arabinoxylans), lignin, and pectin [138]. The positive effects of dietary fiber, such as prebiotics, on health are increasingly recognized. Primary and secondary metabolites, which are produced during the fermentation of the dietary fiber by the gut microbiota, were previously correlated with many health benefits. Several studies found in the literature report the beneficial effects of these nutrients against several chronic diseases, such as type 2 diabetes, hypertension, and Crohn’s disease [139,140].

Each year, about 2 million kg of brewers’ spent grains are produced during the manufacturing of beer. This material, which is mostly represented by spent malted barley, is a valuable by-product that retains protein, fiber, and antioxidant compounds [141]. In recent years, an upcycled snack company started its business of reducing food waste in the craft brewing industry [115]. This company, using barley left over from the beer-making process, plays an important role in its community in minimizing waste and reducing reliance on new resources.

In 2020, a food company based in San Francisco (CA, USA) developed and patented an innovative process based on intermittent infrared drying to obtain concentrated flour from sprouted rye and barley, which is generated after the brewing process. This upcycled ingredient solution offers an amount of dietary fiber and proteins three and two times greater, respectively, than conventional wheat flour [116].

For the upcycled products to be considered healthier than other snacks, they often require replacements for key ingredients, such as flour, in order to incorporate more nutritious ingredients. Often, these modifications present a complex manufacturing challenge in producing healthy snacks a way that does not sacrifice taste and texture. The enrichment with plant-based by-products causes some changes in the physical and organoleptic properties of bakery products; thus, the dose of use is a factor that needs to be considered. Indeed, it was reported in some studies that substitution rates above 10% cause a decrease in the sensory properties of the product [142,143]. Actual modern technology and investment in human capital can help alleviate that situation.

In the last few years, plant-based alternatives to dairy milk gained market share and popularity as an alternative to cow’s milk in Europe and the United States. Among the various plant-based alternatives, oat milk quickly became the plant-based milk of choice, recording a global market size of approximately 2.23 billion dollars (US) in 2020 [144]. In the context of the circular economy, as often reported in the literature, oat by-products have a favorable protein composition and provide other valuable nutrients for human consumption, such as dietary fiber, especially β-glucan, which remain unexploited in the soaking process [145]. Moreover, several patents developed innovative production processes, and many companies around the world started to commercialize oat flour made from oat pulp, also known as oat okara, which is leftover during milk production. Typically, oat-processed material comprises 10–45% proteins and 10–50% dietary fibers [117,118,119]. 

Another important part of the human diet is meat, poultry, and fish. They provide proteins with high biological value, fats, minerals, and vitamins [146]. On the other hand, they are perishable materials because of their high water, lipid, and protein content [147]. Losses of such commodities are usually due to lipid oxidation and microbial spoilage. Using natural preservatives instead of synthetic ones is a sustainable and innovative way of hindering spoilage processes in meat, especially when processed. 

In recent years, a Dutch food and biochemical company was granted a European patent for an advanced fermentation technology that uses agricultural by-products, which allows the firm to produce antimicrobial and antifungal organic acids. The company developed a process that enables the production of lactic and acetic acids, which can be used to extend the shelf-lives of meat and bakery products. The process patented in 2020 involves the fermentation of plant-based by-products, such as pears, apples, pumpkins, and tomatoes, with selected food-grade cultures [120]. According to several studies, the resulting metabolites represented by organic acids, such as acetic, lactic, and propionic acids, have distinctive roles in food preservation [148,149].

In addition to food applications, fruit and vegetable by-products can be used as nutraceuticals due to the health-enhancing substances they contain [150]. The term “nutraceuticals”, which is a hybrid of the words “nutrition” and “pharmaceutical”, refers to a non-toxic extract supplements with scientifically proven health benefits for both disease treatment and prevention [151]. Probiotics, prebiotics, antioxidants, polyunsaturated fatty acids, etc. are the most common food components used as nutraceuticals [152]. In this context, fruit and vegetable by-products are also rich sources of bioactive phytochemicals [153].

It was reported that lycopene extracted from tomato pomace and some phenolic compounds obtained from grape seed extract are valuable compounds due to their health effects [154,155]. In this regard, a few patents were previously published relating to the use of fruit and vegetable by-products as a source of nutraceutical compounds. In 2020, an American wine producer, working in collaboration with the University of California Davis (UC Davis), patented a natural product based on Chardonnay grape seeds that remain after the pressing process [121]. The researchers demonstrated that the Chardonnay grape by-product, compared to other grapes, contains abundant levels of flavan-3-ols, and 36 distinct oligosaccharides were discovered. Through modulation of the intestinal microbiota and its metabolic processes, the bioactivity of these compounds induces an amelioration in insulin resistance, hepatic lipid accumulation, and glucose production [156,157].

A Chinese manufacturer patented an innovative process to produce dietary supplements containing lycopene from tomato by-products [122]. The company registered a patent for the extraction method, which offers the health benefits of the antioxidant through a technology that removes pesticides and increases the quality of lycopene.

### 4.2. Cosmetic Application

Many natural cosmetics brands use ingredients extracted from food industry waste. Ever-more companies focus on creating cosmetics with reduced environmental impacts that are made of ingredients that would otherwise be considered waste, such as by-products from the juicing industry, seeds, fruit waters, cortexes, leaves, pomaces, stones, and peels. Industrial plant-derived food waste could represent, for the cosmetic field, an important natural source of biodegradable, skin-friendly, and environmentally friendly ingredients, such as proteins, polysaccharides, lipids, fibers, vitamins, phytochemicals, and minerals, that can exert either functional or technical activities, including antioxidants, moisturizing, nourishing, antiwrinkle properties, thickening, stabilizing, etc. [158,159,160]. 

Among plants, agaves represent a group of succulent plants with wide applications that generate different by-products. In particular, the decortication of leaves led to the massive production of hard fibers rich in polysaccharides and fructooligosaccharides, which can be employed in the cosmetic industry [161]. Barreto et al. obtained from the industrial by-product of *Agave sisalana* a polysaccharide-enriched fraction and used it to develop cosmetic nanoemulsions for the improvement of skin moisturization [162]. In a recent patent, extracts of *Agave tequilana* were proposed for hair care applications [123]. Different formulations showed activity in the promotion of the stimulation of blood vessels and the migration of endothelial cells, thus enhancing the growth of the hair follicle. The promotion of the shine and the appearance of the hair and scalp was observed due to the renewal of epidermal keratinocytes.

A company in Martinique patented a process to convert the actives extracted from the skin and pulp of green, yellow, and pink bananas, which were part of *Musaceae* family, into upcycled cosmetic ingredients. These molecules, being rich in phytosterols, polyphenols, and polyunsaturated fatty acids, were formulated and are still being sold in different skincare and lip care products, such as exfoliants, toners, creams, face oils, lip balms, etc., that are characterized by firming, anti-aging, and brightening properties [124].

*Mangifera indica* L., which is commonly known as mango, is a fruit plant extensively grown in tropical areas. Being a very popular fruit in the world market, large quantities of waste are associated with its processing [163]. Overall, 30–50% of the fruit is disposed of, mainly without treatment, or becomes feed for animals. Mango waste consists of peels and seeds: the former product represents 7% to 24% of the fruit’s mass, while the latter is 20% to 60% of the fruit’s mass. Similarly, the kernel inside the seed is about 45% to 85% of the seed’s weight. Depending on the variety and method of extraction applied, mango seed kernels contain the following antioxidant biomolecules: carotenoids, tocopherols, and phenolic compounds, such as catechin, quercetin, anthocyanins, and gallic, caffeic, ellagic, and ferulic acids, among others. Mangiferin is a phenolic compound, which is known as a super antioxidant, present also in mango peels and leaves: its anti-collagenase, anti-elastase, and anti-tyrosinase activities were described in a patent [125] used to create anti-aging cosmetics and dermatological products (e.g., depigmenting).

In addition to antioxidants, the kernel contains approximately 34% starch, of which about 20% is highly rich in tannin [164], which is an anti-nutritional compound that could be exploited in the cosmetic field as a thickening and stabilizing ingredient and a gelling and water retention agent, as well as for the preparation of coatings films for the controlled release of functional ingredients [165]. Proximate analysis of mango kernel reveals an amount of fat ranging from about 8% to 25% [163]. The dominant fatty acids in mango butter, which are obtained through cold pressing from kernels, are stearic and oleic acids (higher than about 38% and 41%, respectively). Kernel butter is considered a multifunctional cosmetic ingredient that can be used in skincare, hair care, and decorative products; due to its high content of unsaponifiable matter, such as vitamins (e.g., tocopherols), sterols, and squalene, kernel butter is characterized by emollient, antioxidant, and antiwrinkle properties. Hence, butter obtained from mango by-products can be suggested as a cheaper and more sustainable alternative to cocoa butter, which, despite a similar physicochemical composition, also shows lesser spreadability. 

The *Olea europaea* L., which is a traditional Mediterranean plant widely employed to produce olive oil, leads to the massive issue of by-products, which are mainly represented by the leaves, olive stones, olive mill wastewater, and olive mill pomace. During industrial oil production, enormous amounts of waste are generated; this waste can be upcycled since plenty of bioactive compounds are useful in the formulation of cosmetic products. In fact, these compounds show high antioxidant properties, characteristic fatty acids profiles, and interesting mineral compositions [166]. The leaves are rich in phenols, which are present in the following great varieties: luteolin, rutin, catechin, vanillin, caffeic acid, and oleuropein—the final variety is the most abundant, possessing free radical scavenging properties, as well as skin protection and anti-aging functions. To improve the cell renewal process and the skin appearance, a French cosmetic company recently proposed an exfoliating powder obtained via grinding and sieving olive stones. This powder represents a bio-based alternative to synthetic scrubs and a raw material that, although considered a by-product, contains an oil rich in vitamin E and oleic and linoleic acids [126]. Pectin polysaccharides and hemicellulosic polymers, which are rich in xylans and present in the olive mill pomace, are widely used as emulsifiers and rheology modifiers in cosmetic products, conferring sensory characteristics. In addition, as reported in a patent and a recent study, the olive mill pomace contains squalene, which is a terpenoid molecule naturally present in human sebum that acts as a skin permeation enhancer and exerts antioxidant activity [127,167]. 

The extraction of *Moringa peregrina* seeds was illustrated in a recent patent [128]. The seeds of Moringa, which is a plant predominantly found in Eastern Africa and the Middle West, are used for the extraction of a widely used oil. The oil production leads to the formation of a cake, which is a by-product used as a nutritional source for animal feed. The patent deals with the upcycling of this cake in the nutraceutical and cosmetic industry due to the presence of numerous bioactive molecules. The seed extract contains a peptide hydrolysate that is useful for the production of vegetable hard or soft capsules for food supplements. The seed cake is also suitable for the development of personal care products, such as detergents; skincare products, such as creams; oil-in-water and water-in-oil emulsions; multiple emulsions; gels, lotions; sticks; or powders. In hair care, the Moringa by-product can be used in shampoo, conditioners, lotions, styling creams, gels, masks, etc. The seed cake can also be exploited in makeup products, such as mascaras. Different cosmetic compositions containing Moringa seed cake were described as natural moisturizers, and the water retention capacity of the skin was confirmed using corneometer measurements. Moreover, a protein obtained via aqueous extraction of the seeds of another species of the Moringaceae, i.e., *Moringa oleifera*, was formulated in various cosmetic compositions, and their softening, moisturizing, and anti-pollution effects on the skin were evaluated.

## 5. Conclusions

Various strategies and regulations are used for tackling climate change, environmental degradation, and economic issues related to food waste generation and its disposal. Taking into consideration the first decision-making level, several initiatives were previously adopted in response to the problem, including the SDGs, the European Green Deal, the Circular Economy Action Plan, and the Farm-to-Fork Strategy. Moreover, the spontaneous participation, voluntary approaches, and economic investments by companies are required to reduce the generation of food waste when possible or to convert it into by-products. According to the actual scientific and patent literature, many by-products become new products through an invention that raises their value above zero. Currently, the upcycling of plant-derived by-products at an industrial level in the cosmetic, food, and nutraceutical industries reveals promising results.

Considering the remarkable potential of the unexplored nutrients and bioactive compounds found in food and agricultural residues, which, in recent years, represented market trends, such as prebiotic oligosaccharides and protein hydrolysates, many nutraceutical, food, and cosmetic applications could be explored. Most of the plant-based by-products mentioned in the previous chapters are rich in dietary fiber and protein, which can represent a valuable raw material for different industries.

In this regard, various scientific and technological challenges need to be addressed to increase the extraction efficiency and boost the bioavailability, not just at lab scale but also at industrial premises, of such value-added substances. These also include the use of green solvents and the implementation of low-energy continuous processes to guarantee a totally sustainable approach from all points of view. This utility could be a strong incentive to find new stakeholders in the business industry to support the utilization of food waste as a new resource.

## Figures and Tables

**Figure 1 foods-12-02183-f001:**
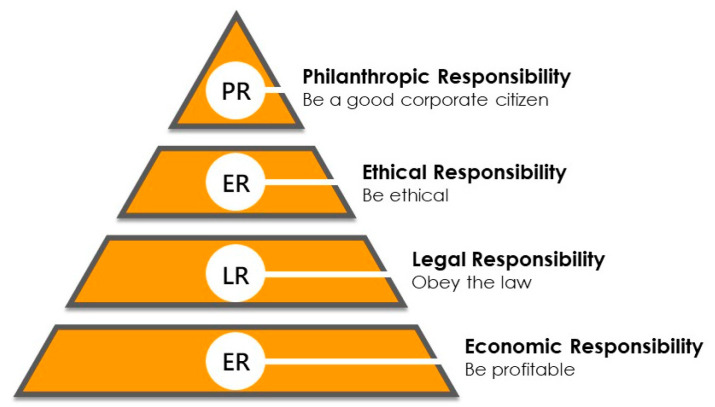
Carroll’s pyramid model of corporate social responsibility.

**Figure 2 foods-12-02183-f002:**
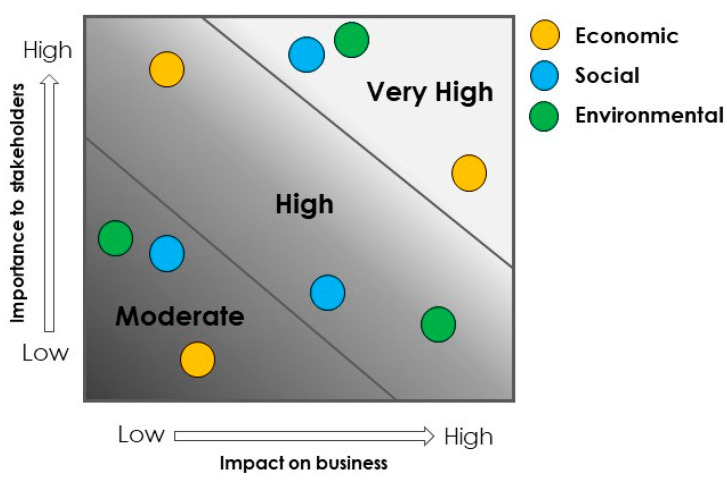
Illustration of a materiality matrix.

**Figure 3 foods-12-02183-f003:**
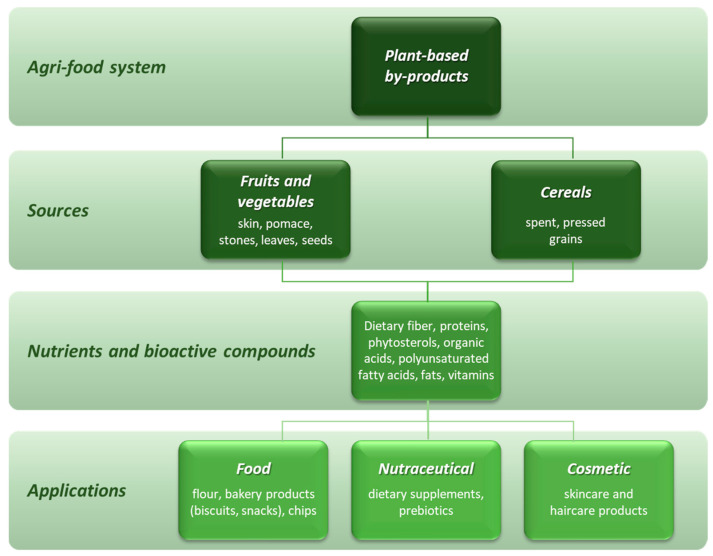
Recovery and applications of high added-value components from plant-based by-products.

**Table 1 foods-12-02183-t001:** Overview of selected patents and industrial applications of plant-based by-products recovered from agri-food wastes.

Source	Value-Added Component	Application	Reference
Carrots, cucumbers, kale, and celery pulp	Dietary fiber and minerals	Snack and chips	[114]
Rye and barley spent grains	Dietary fiber and proteins	Snack and flour	[115,116]
Oat pulp	Dietary fiber and proteins	Flour	[117,118,119]
Pears, apples, pumpkins, and tomatoes pomace	Organic acids	Food preservation	[120]
Chardonnay grape seeds	Oligosaccharides and polyphenols	Dietary supplements	[121]
Tomatoes skin	Lycopene	Dietary supplements	[122]
Agave leaves	Fructooligosaccharides and inulin	Shampoo and hair growth serum	[123]
Skin and pulp of green, yellow, and pink bananas	Phytosterols, polyphenols, and polyunsaturated fatty acids	Skincare and lip care products	[124]
Mango peels and leaves	Mangiferin	Anti-aging cosmetics	[125]
Olive stones and pomace	Vitamin E, oleic and linoleic acids	Exfoliating powder	[126]
Olive mill pomace	Hydroxytyrosol, tyrosol, oleuropein, squalene, and α-tocopherol	Ointments, creams, lotions, and shampoos for dry skins and dermatitis	[127]
Moringa peregrina seed cake	Protein hydrolysates	Cosmetics and nutricosmetics for skin- and hair-care applications	[128]

## Data Availability

The data used to support the findings of this study can be made available by the corresponding author upon request.

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
