# Peer review of "From Industrial Food Waste to Bioactive Ingredients: A Review on the Sustainable Management and Transformation of Plant-Derived Food Waste"

_foods, 2023, doi:10.3390/foods12112183_

Round 1

Reviewer 1 Report

The reviewed work is presenting a good view about obtention of bioactive ingredients from food industrial waste. This topic is very important because it is a intimate relationship with Sustainable Development Goals.

I have only simple suggestions:

(i)             at line 29 to change “ONU´s” by NU.

(ii)           at line 242 to change "Figure 1" by Figure 2,

(iii)         at lines 248-251 to check the edition.

Author Response

Response Letters to the Reviewers

19/05/2023

Foods

foods-2386659

Title: “From industrial food waste to bioactive ingredients: a review on the sustainable management and transformation of plant-derived food waste”.

Section: Special Issue Editorial Board Members’ Collection Series: “Challenges in Maintaining Sustainable Food Systems in Changing Climates”

Dear Respected Editor and Reviewers,

Thanks for your highly insightful and positive comments on our review entitled “From industrial food waste to bioactive ingredients: a review on the sustainable management and transformation of plant-derived food waste” (ID: foods-2386659). The constructive comments helped us to increase the quality of our manuscript. Upon your concern and comments, we have carefully revised the manuscript. All amendments made on the revised manuscript are marked and the detailed information of the responses to the comments is presented as follows.

We hope that the revisions in our manuscript will be enough to make this review suitable for publication in Foods. If there are any further comments/suggestions, please inform me any time. Thank you again for your kind help.

Best regards,

Prof. Dr. Fatih Oz

Response to the Reviewers' comments:

Reviewer #1:

The reviewed work is presenting a good view about obtention of bioactive ingredients from food industrial waste. This topic is very important because it is a intimate relationship with Sustainable Development Goals.

Thank you for your insightful remarks on our paper. We'd like to express our deepest gratitude to Reviewer 1 for his/her thoughtful and useful comments. We would also want to thank the referee for deciding that our study was suitable for publication. We changed the appropriate parts of the paper based on his/her recommendations. Below are our responses to Reviewer 1's questions and remarks.

  1. Introduction

at line 29 to change “ONU´s” by NU.

Thank you for your comment. The abbreviation has been corrected.

  1. Sustainability and food waste: a literature review

at line 242 to change "Figure 1" by Figure 2

Thank you for your comment. The number has been corrected.

at lines 248-251 to check the edition

Thank you for your comment. It has been checked.

Reviewer 2 Report

The paper by Jouhari et al. entitled 'From industrial food waste to bioactive ingredients: a review on the sustainable management and transformation of plant derived food waste' is dealing with an important topic which is agrifood waste sustainable mangement

In my opinion I have some comments to share with authors

1/Abstract section lacks some numerical data related to the degree of management in different countries (developing/industrialized)

2/Introduction section needs to be reorganized and and gathering scetions and paragraphs dealing with the same topics together and using clear sentences

2.1. Sustainable development: the length of this section can be reduced in my opinion 

2.2. Corporate Social Responsibility section should also integrates conclusing remarks by authors and not only merely pure bibliography

2.3.2. Targets, Key Performance Indicators and Reporting: the length of this section can be reduced 

3. Current European strategy and legislation on food waste and by-product : this section contains too much details and can be summarized as well

4.1. Food & nutraceutical application and 4.2. Cosmetic application sections lacks a figure summarizing figures to me more clear for readers

5. Conclusion section is fine

the english languague used is clear

Author Response

Response Letters to the Reviewers

19/05/2023

Foods

foods-2386659

Title: “From industrial food waste to bioactive ingredients: a review on the sustainable management and transformation of plant-derived food waste”.

Section: Special Issue Editorial Board Members’ Collection Series: “Challenges in Maintaining Sustainable Food Systems in Changing Climates”

Dear Respected Editor and Reviewers,

Thanks for your highly insightful and positive comments on our review entitled “From industrial food waste to bioactive ingredients: a review on the sustainable management and transformation of plant-derived food waste” (ID: foods-2386659). The constructive comments helped us to increase the quality of our manuscript. Upon your concern and comments, we have carefully revised the manuscript. All amendments made on the revised manuscript are marked and the detailed information of the responses to the comments is presented as follows.

We hope that the revisions in our manuscript will be enough to make this review suitable for publication in Foods. If there are any further comments/suggestions, please inform me any time. Thank you again for your kind help.

Best regards,

Prof. Dr. Fatih Oz

Response to the Reviewers' comments:

Reviewer #2:

The paper by Jouhari et al. entitled 'From industrial food waste to bioactive ingredients: a review on the sustainable management and transformation of plant derived food waste' is dealing with an important topic which is agrifood waste sustainable mangement.

Thank you for your insightful remarks on our paper. We'd like to express our deepest gratitude to Reviewer 2 for his/her thoughtful and useful comments. We would also want to thank the referee for deciding that our study was suitable for publication. We changed the appropriate parts of the paper based on his/her recommendations. Below are our responses to Reviewer 2's questions and remarks.

  1. Abstract/Introduction

1/Abstract section lacks some numerical data related to the degree of management in different countries (developing/industrialized).

Thank you for your comment. It has been integrated. We explored the literature for information on waste management in both industrialized and developing countries. Unfortunately, the information is rather restricted; nonetheless, we were able to find a reliable source about this theme in line 71 (Introduction) (reference [12]).

  1. Sustainability and food waste: a literature review

2/Introduction section needs to be reorganized and and gathering scetions and paragraphs dealing with the same topics together and using clear sentences

Thank you for your comment. We have been corrected a mistake (line 29) and checked a sentence to make it more fluent (line 83).

2.1. Sustainable development: the length of this section can be reduced in my opinion

Thank you for your comment. For the authors all the content is critical for comprehending the issue in general.

2.2. Corporate Social Responsibility section should also integrates conclusing remarks by authors and not only merely pure bibliography

Thank you for your comment. It has been integrated (lines 222-226).

2.3.2. Targets, Key Performance Indicators and Reporting: the length of this section can be reduced

Thank you for your comment. For the authors all the content is critical for comprehending the issue in general.

  1. Current European strategy and legislation on food waste and by-product

  1. Current European strategy and legislation on food waste and by-product : this section contains too much details and can be summarized as well

Thank you for your comment. For the authors all the content is critical for comprehending the issue in general.

  1. Recent patents and industrial valorization of vegetable by-products for food, nutraceutical, and cosmetic applications

4.1. Food & nutraceutical application and 4.2. Cosmetic application sections lacks a figure summarizing figures to me more clear for readers

Thank you for your comment. It has been added a summary figure (line 593)

Reviewer 3 Report

Dear authors, the review article submitted for review is very interesting. It comprehensively presents both the legal conditions, the current situation with waste and its management (described in the literature).
The in-depth legal analysis leaves nothing to add, it is comprehensive and the reader can easily find his way through the provisions and requirements. In addition, the definitions provided support the situational picture needed to adopt the legal demands.
The next section show the impact of waste management on the social, economic and envirnomental parts. They are correctly described and the relationships well presented.
The last part, in my opinion, could be more elaborate, as it is the key element of the whole article. I understand that there was little information available on this subject - then perhaps the authors would be tempted to include a subsection on suggested/potential future directions of waste management?

Author Response

Response Letters to the Reviewers

Dear Editor, 19/05/2023

Foods

foods-2386659

Title: “From industrial food waste to bioactive ingredients: a review on the sustainable management and transformation of plant-derived food waste”.

Section: Special Issue Editorial Board Members’ Collection Series: “Challenges in Maintaining Sustainable Food Systems in Changing Climates”

Dear Respected Editor and Reviewers,

Thanks for your highly insightful and positive comments on our review entitled “From industrial food waste to bioactive ingredients: a review on the sustainable management and transformation of plant-derived food waste” (ID: foods-2386659). The constructive comments helped us to increase the quality of our manuscript. Upon your concern and comments, we have carefully revised the manuscript. All amendments made on the revised manuscript are marked and the detailed information of the responses to the comments is presented as follows.

We hope that the revisions in our manuscript will be enough to make this review suitable for publication in Foods. If there are any further comments/suggestions, please inform me any time. Thank you again for your kind help.

Best regards,

Prof. Dr. Fatih Oz

Response to the Reviewers' comments:

Reviewer #3:

Dear authors, the review article submitted for review is very interesting. It comprehensively presents both the legal conditions, the current situation with waste and its management (described in the literature).

The in-depth legal analysis leaves nothing to add, it is comprehensive and the reader can easily find his way through the provisions and requirements. In addition, the definitions provided support the situational picture needed to adopt the legal demands.

The next section show the impact of waste management on the social, economic and envirnomental parts. They are correctly described and the relationships well presented.

Thank you for your insightful remarks on our paper. We'd like to express our deepest gratitude to Reviewer 3 for his/her thoughtful and useful comments. We would also want to thank the referee for deciding that our study was suitable for publication. We changed the appropriate parts of the paper based on his/her recommendations. Below are our responses to Reviewer 3's questions and remarks.

  1. Conclusion

The last part, in my opinion, could be more elaborate, as it is the key element of the whole article. I understand that there was little information available on this subject - then perhaps the authors would be tempted to include a subsection on suggested/potential future directions of waste management?

Thank you for your comment. A brief description has been added: we integrated a comment regarding the potential future directions and trends of waste management in Chapter 5 (lines 841-851)
